# Second and Third Generational Advances in Therapies of the Immune-Mediated Kidney Diseases in Children and Adolescents

**DOI:** 10.3390/children9040536

**Published:** 2022-04-11

**Authors:** Ryszard Grenda, Łukasz Obrycki

**Affiliations:** Department of Nephrology, Kidney Transplantation and Hypertension, The Children’s Memorial Health Institute, 04-730 Warsaw, Poland; l.obrycki@ipczd.pl

**Keywords:** immune mediated kidney diseases, evolutions of therapies, biologic drugs, local mechanisms of immunosuppressive drugs

## Abstract

Therapy of immune-mediated kidney diseases has evolved during recent decades from the non-specific use of corticosteroids and antiproliferative agents (like cyclophosphamide or azathioprine), towards the use of more specific drugs with measurable pharmacokinetics, like calcineurin inhibitors (cyclosporine A and tacrolimus) and mycophenolate mofetil, to the treatment with biologic drugs targeting detailed specific receptors, like rituximab, eculizumab or abatacept. Moreover, the data coming from a molecular science revealed that several drugs, which have been previously used exclusively to modify the upregulated adaptive immune system, may also exert a local effect on the kidney microstructure and ameliorate the functional instability of podocytes, reducing the leak of protein into the urinary space. The innate immune system also became a target of new therapies, as its specific role in different kidney diseases has been de novo defined. Current therapy of several immune kidney diseases may now be personalized, based on the detailed diagnostic procedures, including molecular tests. However, in most cases there is still a space for standard therapies based on variable protocols including usage of steroids with the steroid-sparing agents. They are used as a first-line treatment, while modern biologic agents are selected as further steps in cases of lack of the efficacy or toxicity of the basic therapies. In several clinical settings, the biologic drugs are effective as the add-on therapy.

## 1. Introduction

There are a variety of immune-mediated kidney diseases diagnosed in children from early childhood to adolescence. Most of them are presented as one of the clinical forms of isolated glomerulonephritis, while the other ones as a vasculitis-related nephritis or a systemic disease with renal involvement. Nevertheless, the clinical pattern, the innate and/or adaptive immune system is upregulated in most cases and several specific mechanisms of immune dysfunction are involved, as an underlying cause. The typical clinical characteristics of nephrotic glomerulopathies are related to the frequent relapses and the remission dependence from several drugs (as the dose reduction or drug withdrawal results in a relapse of the disease). The hypocomplementemic kidney diseases are often resistant to the standard therapies. The scientific knowledge and clinical experience regarding the relevant therapies have evolved over time. The adaptive immune system was a general target of steroids, which has been the first-line therapy for decades [1,2,3,4,5]. The steroid-related toxicity, especially important in the long-term treatment, was a reason for introducing the steroid-sparing, primarily non-selective antiproliferative drugs, including cyclophosphamide (CYC) and azathioprine (AZA) [5,6,7,8,9]. Treatment with these drugs was not routinely followed by therapeutic drug monitoring (TDM), which precisely evaluates the exposure to the native drugs or its active metabolites This monitoring possibility has appeared more recently (for CYC and AZA) [10], however, it is not frequently used in clinical practice. The next step in specific pharmacotherapy was related to the wider availability of calcineurine inhibitors—cyclosporine A (CsA) and tacrolimus (TAC) [11,12,13,14,15]—and the selective antiproliferative drug mycophenolate mofetil (MMF) [16,17]. The exposure to all of them is easily measurable with TDM. The next step of progress was associated with introduction of the biologic drugs, targeting specific receptors or narrow pathways of the innate and/or adaptive immune system, important for the variable mechanisms of a specific kidney disease [18,19,20,21]. The final issue in the therapy of immune-mediated kidney diseases was the discovery of local mechanisms of specific drugs, targeting podocytes in glomeruli, independently of the impact on the immune system [22,23]. Evolution of the management of pediatric immune-mediated kidney diseases is presented in Figure 1.

The variety of drugs allows individualization of therapy; however, from a clinical point of view, there are several significant limitations, which may decrease the efficacy and overall suitability of a certain protocol in a particular case. The summary characteristics of the classic drugs used in pediatric immune-mediated kidney diseases is presented in Table 1, and of the modern biologic drugs in Table 2.

## 2. Therapeutic Drug Monitoring (TDM) and Immunomonitoring in Kidney Diseases

TDM is a mandatory tool in clinical management of immune-mediated kidney diseases. The general rules of TDM are summarized in Table 3. There are two distinct ways of surveillance of the exposure to the drugs, used in the therapy of the kidney diseases: a classic TDM, based on a regular monitoring of drug concentration in the blood (or plasma), as by a single test (pre-dose trough level; C_0_) or by the parameter defined as the area under the curve (AUC), combining the results of multiple tests/daily [40,41,42] and immunomonitoring of the number of target cells, depleted by a specific monoclonal antibody (like B CD_20_) (pre-dose test) [43]. The associations between the dose and drug concentration or a number of target cells are clinically relevant in terms of systemic action of the certain drugs; however, in terms of local effect on podocytes, the associations are not clear. The clinical rules of TDM and immunomonitoring are presented in Table 3.

## 3. Nephrotic Syndrome—An Alternative Management beyond the Classic Protocols

Nephrotic syndrome is an “umbrella term” covering the typical “idiopathic” nephrotic syndrome in children, with minimal change and (less frequently) focal segmental glomerulosclerosis (FSGS) in a kidney biopsy, as well as symptomatic proteinuria presented in other types of glomerulonephritis (GN), especially in mesangial-proliferative GN and membranous nephropathy. The pediatric nephrotic “family” is heterogeneous in terms of underlying immune-mediated (non-genetic) mechanisms, and this was also important for selection of the secondary-step therapies. One of these mechanisms is related to the dysregulation of the interaction between B and T cells, resulting in their dysfunction. B and T cells may produce several bioactive cytokines, which are regarded as “protein permeability factors”, the heterogeneous family of cytokines binding specific targets on podocytes, impact their functionality and shape, and also provoke and maintain the clinically overt proteinuria. This family includes cardiothropin-like cytokine-1 (CLC-1)—a soluble urokinase-type plasminogen activator receptor (suPAR), IL-4, IL-10, IL-13, TNFα and anti-CD_40_ antibody (and probably many others, still not defined) [44]. Some of these processes may be modified by the use of rituximab, a IgG1 kappa- type monoclonal antibody, which induces lysis of B cells (pre-B, immature, mature and activated), expressing the CD_20_ receptor. The effect of rituximab is exerted via B cell depletion, indirect impact on T cells and local activity in podocytes. The depletion is executed by the antibody-dependent or complement-dependent cytotoxicity resulting in apoptosis, inhibition of proliferation and alteration of the cell cycle. Activation and differentiation of T cells, as well as the ability to release several podocyte-specific cytokines are (in some part) regulated by B cells, which produce specific costimulatory signals; therefore, B cell depletion may ameliorate the T-cell specific activities. Rituximab is also able to modulate the T_reg_ and B_reg_ cell interactions, which are involved in the mechanism of a relapsing nephrotic syndrome [24,25,45]. Rituximab also exerts a specific local effect on podocytes (independently from its effect on B CD_20_ receptor), related to the stabilization of the podocyte’s cytoskeleton by regulation (preservation) of sphingomyelin phosphodiesterase acid-like 3b (SMLPD-3b)—a protein involved in the podocyte cytoskeleton activity regulation [23]. Clinical relevance of this mechanism is not clear [46]; however, it is speculated that partial remissions achieved in some of the treated patients or the lack of correlation between number of peripheral CD_20_ cells (or CD_19_; used for drug effect monitoring) and clinical efficacy (seen also in some patients) are the results of the local (not systemic) effect of rituximab [26]. Whether the routine of the dosing regimen used for systemic treatment (e.g., 4 × 375 mg/m^2^ of BSA) is also adequate for local effect exerted by rituximab is not clear either.

## 4. Cyclosporine A as a Podocyte Cytoskeleton Stabilizer

The systemic immunosuppressive mechanism of cyclosporine A (CsA) is related to the inhibition of the nuclear factor of activated T cells (NFAT) signaling in the T lymphocytes. There is also another distinct mechanism of CsA (not dependent on NFAT inhibition in T cells), which is related to the stabilization of the actin cytoskeleton in the podocytes. CsA blocks the calcineurin-mediated dephosphorylation of synaptopodin in podocytes, preserving the phosphorylation-dependent synaptopodin–14-3-3β interaction. This protects synaptopodin from cathepsin L-mediated degradation [47]. In clinical translation, it may reduce the degree of proteinuria, when a nephrotic patient is constantly exposed to cyclosporine A [48]. The exact association between the value of CsA concentration in the blood and activity of the local mechanism in terms of the podocyte stabilization in unclear, however there are patients with genetic forms of nephrotic syndrome, where a major underlying cause of proteinuria is a defect of kidney microstructure, who gain a clinical profit (decrease of proteinuria during constant treatment) under the routine doses and the trough concentrations of the drug [49,50]. This suggests that a routine “clinical” dosing is relevant also for the local effect of CsA.

## 5. Is Targeting of a CD80 Molecule-Clinically Relevant?

The CD_80_ (B7-1) is a molecule expressed on the surface of T cells, dendritic cells and podocytes. The “two-hit” hypothesis suggests that an immune response to external trigger (e.g., viral infection) in the immunocompetent humans is limited to a transient expression of CD_80_ on podocytes, which may induce a short-term and minor proteinuria. Humans with a sustained dysfunction of the regulatory T cells (T_reg_) react to the exposure to the similar trigger with marked and sustained expression of CD_80_ in podocytes, which causes a dysregulation of cytoskeleton and in clinical consequence—a persistent heavy proteinuria (nephrotic syndrome) [51,52]. CD_80_ is also expressed on the T cells infiltrating the kidney tissue (at least) in some nephrotic children [53]. CTLA-4 (cytotoxic T-lymphocyte-associated protein 4), expressed on podocytes and T_regs_, is an important co-factor in this mechanism [54]. High urine excretion of CD_80_ is being regarded as a biomarker of minimal change disease [55,56]. Translation of these mechanisms to the potentially relevant therapy involves abatacept, which is a fusion molecule including the Fc region of IgG1 and cytotoxic T-lymphocyte-associated protein 4 (CTLA-4-Ig). The drug acts as a co-stimulatory inhibitor and targets B7-1 (CD80) and CD86, disrupting the activation of T-cells. It also blocks the disruption of binding of talin to ẞ1-integrin caused by CD80. This mechanism should ameliorate a podocyte injury and decrease proteinuria. A primary clinical report showed clinical efficacy of abatacept in 4 (of 5) cases (including one child and two adolescents), resistant to other drugs, including rituximab [57]. Further reports were more reluctant in terms of the efficacy of abatacept [27,58]. Overall, the clinical efficacy was reported in about 44% of nephrotic patients [27]. Probably, the use of abatacept should be limited to the cases, in which the renal biopsy (of native or transplanted kidney) will confirm the local expression of B7-1 (CD_80_) in podocytes and/or in which the baseline value of the urinary excretion of CD_80_ is markedly increased [59].

## 6. The Role of Angiopoietin-like-4 (Angptl4) in Minimal Change Nephrotic Syndrome and a Hypothetic Target-Treatment

Podocytes of patients with minimal change nephrotic syndrome (MCNS) express angiopoietin-like 4 (Angptl4). There two types of Angptl4 protein: hyposialylated Angptl4 secreted locally by podocytes, and a neutral form, present in high concentration in the plasma of nephrotic patients. The podocyte-derived hyposialylated Angptl4 mediates proteinuria in patients with MCNS and binds to a glomerular basement membrane and endothelial cells [60,61]. Converting a hyposialylated Angptl4 to the sialylated form with N-acetyl-D-mannosamine (a precursor of sialic acid), that can be taken up and stored in podocytes, may reduce proteinuria and has the potential to be used in small maintenance doses to prevent the relapses of MCNS [62]. Despite the attractive hypothesis, there are no available relevant clinical data so far.

## 7. Complement System Blocking Agents in Glomerular Diseases (with Defect of Complement)

The role of a complement system in the mechanism of the specific glomerular kidney diseases is very complex. In physiology, the complement is being used as a natural tool for the clearance of immune complexes; however, it may also act as a link between innate and adaptive immunity, activating B-cells and T-cells in the local renal environment. The genetic mutations of complement components and/or regulators or a presence of specific acquired autoantibodies targeting the complement components lead to the excessive activation of the alternative pathway and to the glomerular deposition of the complement fragments (debris), as may be documented in the kidney biopsies [63,64,65]. The clinical spectrum of complement-associated kidney diseases includes membranous nephropathy, lupus nephritis, C_3_ glomerulonephritis (C_3_GN), membranoproliferative glomerulopathy type I, dense deposit disease (DDD), IgAN and ANCA-associated vasculitis. Several monoclonal antibodies and other molecules targeting the pathways of complement system are currently evaluated in clinical trials associated with different types of glomerulonephritis [28]. The pediatric experience is limited to the use of eculizumab in cases of C_3_GN, dense deposit disease and membranoproliferative GN [29,30,31]. The most significant experience was reported in the multicentre series of children and adolescents with C_3_GN, treated with eculizumab, in whom the efficacy of this monoclonal antibody was confirmed in cases of crescentic rapidly progressive C3 glomerulopathy, while the clinical benefit was limited in the clinically mild cases [32].

## 8. Biologic Drugs in Pediatric Systemic Vasculitis

Systemic vasculitis is a family of diseases characterized by the presence of vascular wall inflammation and involving multiorgan symptoms. Despite the low incidence in children (mainly adolescents), this group of diseases, and particularly ANCA-associated vasculitis (AAV)), has gained an access to several biologic agents, targeting the underlying mechanisms. The pathogenesis of AAV is multifactorial and genetic/epigenetic factors interact with different external triggers. Dysregulation of B cells and lack of balance between T helpers and T effectors lead to the production of ANCA (anti-neutrophil cytoplasmic antibody), activation of neutrophils, and further damage of vessel walls with the late multiorgan consequences [66]. The relevant biologic agents, preliminarily described in case reports, then evaluated in several clinical trials enrolling adult and (in some projects) also pediatric and adolescent patients, include rituximab (anti-B CD_20_ moab), infliximab (anti-TNFα moab), etanercept (TNFα-receptor blocker), abatacept (CTLA-4 Ig Fc fusion protein), alemtuzumab (humanized anti-CD52 moab) and tocilizumab (anti-IL6 moab) [67]. The use of rituximab is aimed to block B-cell dependent T-cell activation, causing the overproduction of several interleukins (like IL-5). A pediatric trial (among several others enrolling only adult patients), in which children with granulomatosis with polyangiitis (GPA) and microscopic polyangiitis (MPA) received four doses of rituximab, has demonstrated a positive effect, as the remission (verified with the Pediatric Vasculitis Activity Score) was achieved in 56%, 92%, and 100% of patients at months 6, 12, and 18, respectively [33]. The idea of using TNFα blockers is based on the mechanism, in which inhibition of TNFα decreases the formation of granuloma, as a result of the TNFα-mediated activation of neutrophils that enhances the ability of ANCA to stimulate degranulation of neutrophils, is a which important factor in a process of the vascular wall damage [67]. The study named WGET (Wegener’s Granulomatosis Etanercept Trial), which enrolled adult patients, did not prove a clinical benefit in terms of achieving remission, as compared with the placebo group (notably, both groups also received standard care drugs, including steroid and cyclophosphamide or methotrexate [34]. Abatacept was used as a T-cell activation blocking agent in adult patients with GPA (granulomatosis with polyangiitis). Overall, 80% of enrolled patients achieved remission, and in 78% of cases a complete steroid withdrawal was possible [35]. Alemtuzumab, a depleting moab against CD52, the receptor expressed by the several cells (T lymphocytes, monocytes, macrophages), was used in adult AAV patients. The remission was achieved in two thirds of patients at six months, and was maintained to twelve months in one third of them [36].

The clinical view on the real success rate in several trials, comparing the efficacy and safety of modern biologic agents with “traditional” drugs, is complex, as in several cases the interpretation of final outcome and safety was difficult and not always in favor of the use of particular biologics, which have been used as the add-on therapy.

## 9. Biologic Drugs in Systemic Lupus Erythematosus with Renal Involvement

Several specific mechanisms relevant for the lupus erythematosus with renal involvement have been currently targeted by narrow biologic agents, evaluated in clinical trials. One of them is B-cell activating factor (BAFF). The agents used in this setting included blisibimod (A-623, AMG 623), a fusion protein, built of tetrameric BAFF binding domain fused to human IgG1 Fc region and belimumab (a monoclonal antibody). Clinical data on blisibimod from CHABLIS-SC1, randomized, double-blind, placebo-controlled clinical trial, conducted in adult patients, treated previously with variable drugs (including steroids, MMF, methotrexate, AZA or antimalarian agents), showed that the primary end-point (week 52 SLE Responder Index-6) was not achieved; however, the use of blisibimod was associated with decreased proteinuria, successful reduction of exposure to steroids, and a positive response of the relevant biomarkers [37]. Several clinical trials have been conducted in adults and children with SLE, evaluating the efficacy and safety of belimumab. Overall, in children, belimumab reduced the risk of severe flares by 64% (versus standard therapy), which was more pronounced than in the adult studies (23–50%, respectively) [38]. Atacicept is a recombinant fusion protein consisting of the binding portion of transmembrane activator and CAML interactor (TACI; also known as tumor necrosis factor receptor superfamily member 13B), that binds to BLyS (B lymphocyte stimulator) and APRIL (a proliferation-inducing ligand), inhibiting interactions with their specific receptors. An adult trial with extended follow-up showed a reduction of the incidence of severe SLE flares and prolongation of time to the occurrence of severe flares with a high dose of the drug, as compared to the placebo [39]. Blisibimod and atacitept have also been evaluated in adult patients with IgA nephropathy-related persistent proteinuria >1 g (NCT02062684; NCT02808429); however, (published) results are not available. In general, the role of these kind of narrow-targeted therapies may be regarded mainly as add-on treatments, ameliorating the clinical course of the disease and acting as the steroid-sparing drugs. Anyway, the direction towards the use of “therapeutic arrows” [68], drugs and agents designed based on increasing molecular knowledge on the mechanisms of immune-mediated kidney diseases is a promising clinical approach, with prospects for the future.

## 10. Summary

Upregulation of the innate and/or adaptive immune system leads to the development of a variety of immune-mediated kidney diseases in children.There is an ongoing progress in pharmacotherapy of immune kidney diseases, based on scientific knowledge, which defines detailed, variable underlying disease-related mechanisms.The adaptive immune system is a target of steroids, antiproliferative drugs, calcineurin inhibitors and several receptor-specific biologic agents.The innate immune system is a target of specific monoclonal antibodies.Surveillance of the current therapies is based on therapeutic drug monitoring and/or immunomonitoring.Apart from the effect on the immune system, specific drugs (calcineurin inhibitors, rituximab, abatacept) also exert a local effect on the microstructure of the podocyte cytoskeleton, which may be clinically relevant in selected cases.

## Figures and Tables

**Figure 1 children-09-00536-f001:**
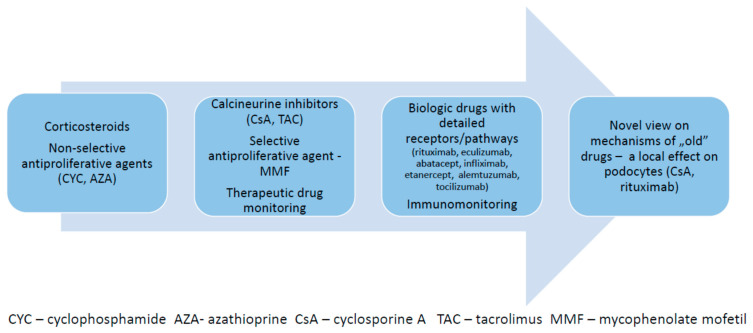
Evolution of therapies used in pediatric immune-mediated kidney diseases.

**Table 1 children-09-00536-t001:** Summary characteristics of the “classic” drugs used in pediatric immune-mediated kidney diseases [1,2,3,4,5,6,7,8,9,10,11,12,13,14,15,16,17].

Drug	Systemic Target/Mechanism	Local (Kidney) Target/Mechanism	Diseases	Generational Line of Therapy
Steroids (including ACTH)	Induce downregulation of pro-inflammatory genes by binding to a specific cytoplasmic glucocorticoid receptor (GR); regulate gene expression by binding to glucocorticoid response elements (GRE) on DNA	Attenuation of podocyte apoptosis Blocking of TRPC6 signal pathway Blocking of CD80 upregulation	Nephrotic syndrome All other types of immune-mediated glomerulonephritis	first
Cyclophosphamide	Exerts cytotoxic effect by cross-linking of strands of DNA and RNA, and by inhibition of protein synthesis	none	Nephrotic syndrome Lupus nephritis Vasculitis with renal involvement IgAN	second
Azathioprine	Acts by the incorporation into replicating DNA; blocks de novo pathway of purine synthesis	none	IgAN AAV	second
Mycophenolate mofetil	Mycophenolic acid (MPA), a drug derivative, acts as a selective, noncompetitive inhibitor of inosine monophosphate dehydrogenase IMPDH); inhibits T- and B-lymphocyte proliferation	Decrease of uPAR expression Modification of ATP depletion (by inhibiting IMPDH) in podocytes	Nephrotic syndrome Lupus nephritis Vasculitis with renal involvement IgAN	second
Cyclosporine A	Inhibits the phosphatase activity of calcineurin, which regulates nuclear translocation and subsequent activation of NFAT transcription factors; CsA is a specific inhibitor of T cell activation	Stabilization of actin cytoskeleton by preserving a phosphorylation-dependent synaptopodin-14-3-3-ẞ integrin interaction Decrease of TRPC6 and ANGPTL4 upregulation	Nephrotic syndrome	second
Tacrolimus	Inhibits T lymphocyte activation and transcription of cytokine genes, including the gene for interleukin-2	Decrease of TRPC6 and ANGPTL4 upregulation	Nephrotic syndrome	third (off-label)

ACTH—adrenocorticotrophic hormone. NFAT—nuclear factor of activated T-cells.

**Table 2 children-09-00536-t002:** Summary characteristics of the biologic drugs used in pediatric immune-mediated kidney diseases [20,21,22,23,24,25,26,27,28,29,30,31,32,33,34,35,36,37,38,39].

Drug	Systemic Target/Mechanism	Local (Kidney) Target/Mechanism	Diseases	Generational Line of Therapy
rituximab	Binds to CD20 on B cells and mediates B-cell lysis and depletion	stabilizes podocytes cytoskeleton by regulation (preservation) of sphingomyelin phosphodiesterase acid-like 3b (SMLPD-3b), a protein participating in the podocyte cytoskeleton activity	Nephrotic syndrome AAV GPA MPA IgA vasculitis Membranous nephropathy	third (off-label) Clinical trials (adults and adolescents) Case reports
eculizumab	Binds to complement protein C5, inhibiting its cleavage to C5a and C5b; this prevents a generation of the terminal complement complex C5b-9	none	Atypical hemolytic uremic syndrome (aHUS) Lupus nephritis C3 nephropathy MPGN type I DDD	first third (off label) second/third (off-label)
abatacept	A fusion molecule including Fc region of IgG1 and cytotoxic T-lymphocyte-associated protein 4 (CTLA-4-Ig) Blocks CTLA-4 present on T_regs_	Blocking of B7-1 signalling and restoration of ẞ1 integrin activation	Nephrotic syndrome GPA (Granulomatosis with Polyangiitis) Takayasu arteritis	fourth (off-label) Clinical trials (adults and adolescents) Case reports
infliximab etanercept	Anti-TNFα moab Human Fc fusion protein- blocks the TNFα -receptor	none	AAV Takayasu disease PAN	Clinical trials (children) Case reports
alemtuzumab	Anti-CD52 depleting moab, lysis of all cells expressing CD52	none	AAV	Clinical trials (adults) Case reports
tocilizumab	Anti-IL6 moab	none	Kawasaki disease	Children (case reports)
anakinra	Anti-IL-1 moab	none	Kawasaki disease	Clinical trial (children)
blisibimod	Blocks of B-cell activating factor (BAFF)	none	SLE	Clinical trial (adults)
belimumab	Anti-BAFF moab	none	SLE	Clinical trials (adults and children)
atacicept	A recombinant human fusion protein that binds to BLyS (B lymphocyte stimulator) and APRIL (a proliferation-inducing ligand); inhibits interactions with their specific receptors	none	SLE	Clinical trial (adults)

AAV—ANCA-associated renal vasculitis. ANGPTL4—angiopoietin-like 4 molecule. APRIL—a proliferation-inducing ligand. BAFF—B-cell activating factor. BlyS—B lymphocyte stimulator. B7-1 (CD80)—a molecule expressed on the T cells, dendritic cells and podocytes surface. CTLA-4 (cytotoxic T-lymphocyte-associated protein-4)—molecular co-factor expressed on podocytes and T_regs_. DDD—dense deposit disease. GPA—granulomatosis with polyangiitis. IMPDH—inosine 5′-monophosphate dehydrogenase. moab—monoclonal antibody. MPA—microscopic polyangiitis. MPGN—membrano-proliferative glomerulonephritis. PAN—polyarteritis nodosa. TRPC6—transient receptor potential channel 6. SLE—systemic lupus erythromatosus.

**Table 3 children-09-00536-t003:** Clinical background of TDM (therapeutic drug monitoring) and immunomonitoring [40,41,42,43].

Rule	Clinical Practice
There is an association between the dose and drug concentration in the blood/plasma.	The activity of drug-specific metabolic pathways is programmed genetically and also is age-dependent.
Pharmacokinetics and pharmacodynamics of the drug depend on function of the specific pathways and organs (routes of drug metabolism and clearance).	There is an association between drug concentration and its clinical efficacy and toxicity.
Specific drugs administered simultaneously may interact, and this reaction changes its’ metabolism and pharmacokinetics.	Multidrug management requires more frequent TDM.
The pharmacokinetic effect of the biologic drug is expressed as a number of targeted cells.	There is an association between the number of target cells and the clinical course of the disease.

## Data Availability

Not applicable.

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
