# Peer review of "Second and Third Generational Advances in Therapies of the Immune-Mediated Kidney Diseases in Children and Adolescents"

_children, 2022, doi:10.3390/children9040536_

Round 1
Reviewer 1 Report
This is a concise and well organized review of an interesting topic. I have made suggestions for minor editorial and English language changes in the attached manuscript. I do think the Title should be changed to "Second and Third Generational Advances in Therapies of Immune-Mediated Kidney Diseases in Children and Adolescents- A Practical Review"

Author Response
Thank you for your comments, suggestions and editing support.
We have modified the title, as was suggested.
We have corrected listed spellings.
Your suggestions helped us to improve the quality of the manuscript.
Authors
Reviewer 2 Report
The review by Grenda and Obrycki is concise and straightforward. I have the following comments.
- The findings related to the effects of drugs/biologics in the tables are chiefly supported by the Review articles. It would be helpful for the potential readers if the authors cite the original publications while reporting the observations specifically related to the effects on the cells in the kidney. Citing the original publications will help in understanding the mechanism if the potential readers want in-depth knowledge about specific drug/biologics.
- The authors are encouraged to also discuss the studies and results of clinical trials (if available) for Blisibimod (A-623, AMG 623), a fusion protein as tested for patients with SLE and IgA nephropathy; Belimumab, a monoclonal antibody against B-cell activating factor (BAFF), and Atacicept, a recombinant fusion protein also against B-cells in their Table 2 and discussion section.
- All the full forms in the table should also be abbreviated in the footnote. For e.g. NFAT also needs to be included in the footnote of Table 1.
Minor:
- There are several grammatical errors including spellings, which must be corrected. Some of the examples of errors in spellings are provided below. Replace these
- “pharmacokinetcs” with “pharmacokinetics”
- “calcineurine” with “calcineurin”
- “signalling” with “signaling”
- “Nephrotic syndrom” with “Nephrotic syndrome”
- “glorulosclerosis” with “glomerulosclerosis”
- “heterogenous” with “heterogeneous”
- “permability” with “permeability”
- “indepedent” with “independent”
- "mulitfactorial" with "multifactorial"
- They need to correct this sentence: “The exact association between a CsA concentration in the blood and the intensivity of the local mechanism in unclear”
- The presentation/formatting of the tables needs to be improved.
- The formatting of references and the style must be improved.
Author Response
Thank you for your comment and suggestions.
We have added several new (original) references, adressed to clinical trials and clinical issues related to the use of particular drugs, listed in the tables. We have added data on the suggested new drugs.
Your suggestions helped us to improve the quality of the manuscript.
Authors